# Shape programming of liquid crystal elastomers
Andraž Rešetič ✉

Liquid crystal elastomers (LCEs) are shape-morphing materials that demonstrate reversible actuation when exposed to external stimuli, such as light or heat. The actuation's complexity depends heavily on the instilled liquid crystal alignment, programmed into the material using various shape-programming processes. As an unavoidable part of LCE synthesis, these also introduce geometrical and output restrictions that dictate the final applicability. Considering LCE's future implementation in real-life applications, it is reasonable to explore these limiting factors. This review offers a brief overview of current shape-programming methods in relation to the challenges of employing LCEs as soft, shape-memory components in future devices.

Liquid crystal elastomers (LCEs) are materials that combine the reversible ordering properties of liquid crystal molecules, i.e., mesogens, with the elastic nature of a cross-linked polymer network. They undergo significant shape changes when exposed to external stimuli such as temperature or light. These external fields disrupt the mesogen orientational order, causing the entire LCE network to reconfigure. This results in material deformation, which returns to its initial shape after the stimuli are removed, with the original network configuration re-established by the elastic and self-ordering nature of the LCE[1].

For these materials to exhibit significant shape changes, a well-defined orientational order of liquid crystal components or mesogens needs to be established with an applied orientational field and imprinted within the material through the crosslinking of the LCE network. This fixation of the mesogen arrangement ensures that the same structure is reestablished after each stimulus, enabling consistent shape changes. LCEs are classified as monodomain when the mesogens are well-ordered across the entire sample; otherwise, they self-assemble into smaller ordered but randomly oriented domains to form a polydomain material. Such polydomain LCEs do not undergo significant shape changes upon activation.

To fabricate shape-morphing LCEs, programming the orientation of the mesogens is an essential step. This process was pioneered by Finkelmann et al. through the introduction of a two-step crosslinking procedure[2]. In this approach, mechanical stress is applied to realign the mesogen components of a partially crosslinked LCE, creating a monodomain LCE network topology. This orientation is then preserved through further crosslinking, resulting in thermomechanical actuation of LCEs. Since then, several new synthesis techniques have emerged for programming shape-changes in LCEs. These include using light-patterned cells for surface alignment, external electric or magnetic fields for 3D control of LC orientation, shear-flow alignment utilized for additive manufacturing, and other innovative

methods. However, existing synthesis methods often limit the size and shape of LCEs, as well as the freedom to control their orientation.

In its basic form, the synthesis of LCEs typically involve three main steps: partial crosslinking to create an isotropic system of orientable polymer chains, followed by the mesogen alignment, and final crosslinking to preserve the aligned LCE network configuration. The synthesis steps can indeed be merged together to provide an in-situ alignment and polymerization, such as in the case of shape programming with surface alignment. Nevertheless, combining different orientational fields and crosslinking methods is shown to be complex, thereby hampering full control over mesogen alignment and geometrical freedom (Fig. 1). As an example, a typical two-step LCE synthesis via thermally induced polymerization often relies on the use of significant amounts of solvents, necessary to ensure the production of a homogeneous mixture and increase the mobility of the reactants, especially during the first partial crosslinking stage. Following the molding process, these solvents must be completely removed prior or after the completion of the final crosslinking reaction, leading to the deformation of the cured LCEs due to deswelling. Furthermore, given that these reactions are conducted at elevated temperatures, an additional thermomechanical relaxation might be expected to occur upon cooling of the aligned specimen. This hampers precise control over the programmed and actuated shapes, especially when aiming for the synthesis of more complex and precisely defined geometries. An alternative approach involves photocuring with UV light[3,4], which can eliminate the need for high temperatures during the synthesis. However, this synthesis process faces constraints due to the limited penetration length of light during curing, which restricts the achievable thickness in the synthesized LCE to several hundred micrometers. It is therefore not possible to use conventional photopolymerization processes to mold LCEs into bulk systems with a thickness comparable to other dimensions, typically in the centimeter range, unless relying on

Jožef Stefan Institute, Solid State Physics Department, Jamova cesta 39, 1000 Ljubljana, Slovenia. ✉e-mail: andraz.resetic@ijs.si

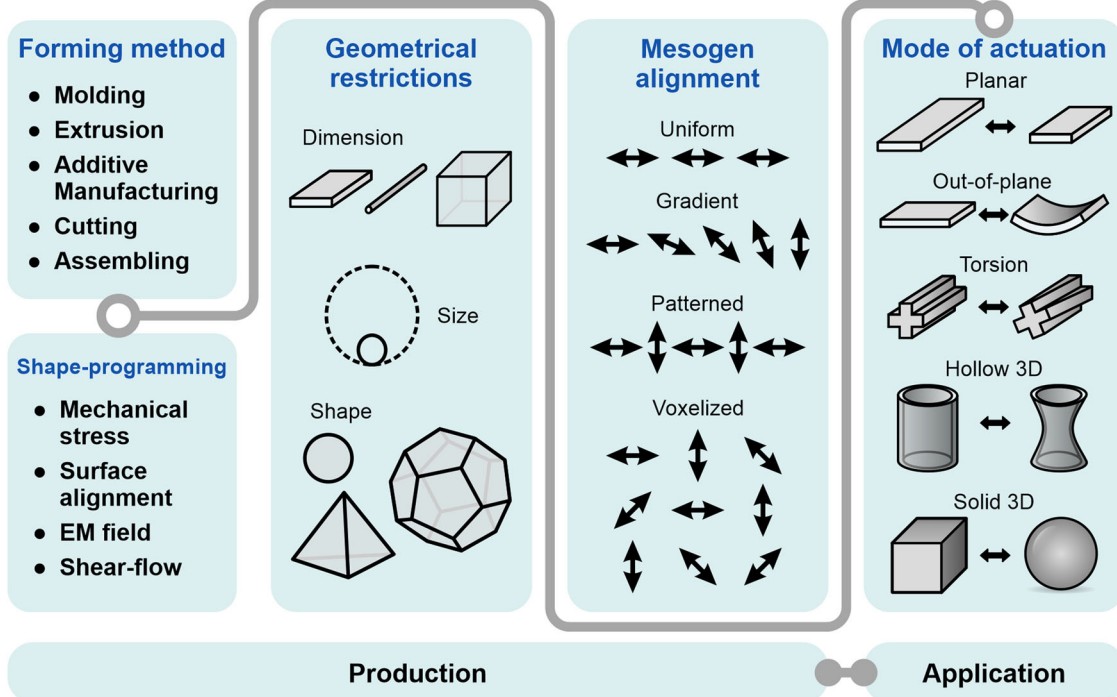

**Fig. 1 | LCE production challenges.** The combination of forming and shape-programming methods is guided by geometrical and mesogen alignment limitations, impacting both material shaping and subsequent shape-change. Each LCE production method permits specific modes of actuation, thereby restricting the potential applications of the final material.

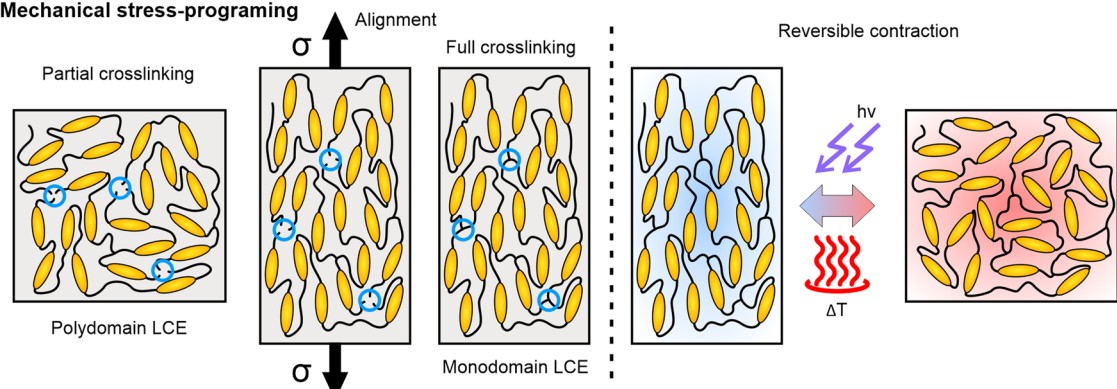

**Fig. 2 | Schematic of basic three-step shape programming method with an external load.** A partially crosslinked LCE is strained and then further crosslinked to preserve the induced monodomain network. With the application of external stimuli, such as illumination with light or increased temperature, the LCE contracts in the direction of the programmed strain.

layering or additive manufacturing techniques. Obstacles such as these greatly hinder the transition of LCEs from a proof-of-concept material to practical application in real-life devices.

This review briefly explores established methods for creating ordered mesogen configurations in LCE systems and discusses the strengths and constraints posed by different shape programming techniques, considering their implementation for large-scale production and applicative potential of LCEs as actuators. The most recent advances that overcome common limitations of more conventional techniques will be presented in greater detail. Highlight is overall given on methods generating sizable specimens within the macroscopic scale, capable of significant deformation and finding potential application as active, shape-morphing mechanical components. The review concludes with an overall discussion and comments regarding the shape programmability and application of LCEs.

## Shape-programming with mechanical stress

The most common shape programming method utilizes the established coupling of the elasticity of the overall polymer network comprising LCEs and the orientational ordering of the mesogens. By reshaping the LCE system, the mesogens are reoriented in the direction of the mechanical stress distributed over the network. The alignment step is performed after the first partial crosslinking, establishing the elastic network for specimen deformation (Fig. 2). The alignment and mechanically induced deformation are then retained after the second crosslinking step, permanently imprinting the ordered configuration into the LCE. After the synthesis, the mechanically programmed LCEs contract along the programmed deformations.

Since mechanical stress is externally applied and exerted over the crosslinked polymer network onto the mesogen components, precise modulation of mesogen alignment over the material's volume is challenging. Rudimentary shape programming is performed by loading the material to

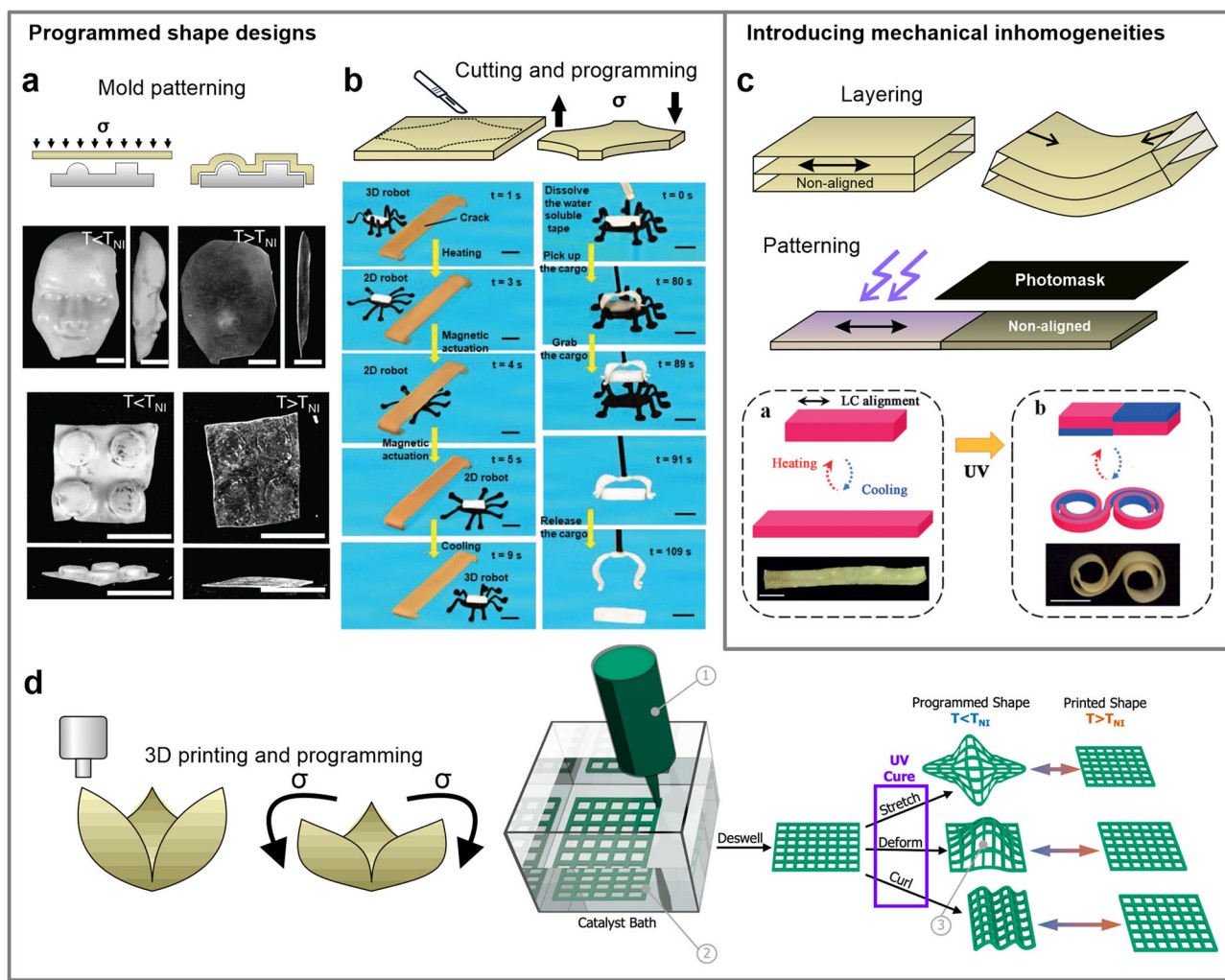

**Fig. 3 | Shape programming techniques based on mechanical stress programming. a** More elaborate programmed shapes are achieved by reshaping LCE sheets around a mold. Scale bars: 10 mm. Reproduced from ref. 11 with permission from the Royal Society of Chemistry. **b** Cutting LCE samples into new geometries and programming specific parts of the sample induces selective shape changes in the same specimen. Scale bars: 5 mm. Reproduced from ref. 12 with permission. © 2021 Wiley-VCH GmbH. **c** Non-linear shape changes can be realized by fabricating layered composites or by patterning the mechanically instilled mesogen alignment. Scale bars: 5 mm. Reproduced from ref. 20 with permission. © 2019 Wiley-VCH Verlag GmbH & Co. KGaA, Weinheim. **d** 3D printing can be employed for fabricating LCEs with complex geometries, ready for further shape-programming. Reprinted with permission from ref. 24. Copyright © 2020 American Chemical Society.

induce elongation, bending, or twisting, resulting in uniform shape changes across the whole specimen between the programmed and initially synthesized shape[5,6], unless relying on targeted or modulated external stimulation[7–9]. Mechanically programmed LCEs can exhibit substantial linear actuations[10], with the magnitude determined by the programming strain, mostly limited by the mechanical strength of the partially crosslinked material after the first step of the synthesis process.

More sophisticated programmed geometries can be realized by applying a non-homogeneous stress field, such as compressing or bending the LCE sheets against a mold[5,11] (Fig. 3a). These LCEs take on the precise shape of the die and return to a flat geometry once heated into the isotropic phase. LCEs can also be intricately cut and exposing different sections of the material to separate shape programming. This leads to the creation of robotic-like devices, such as grippers and walkers[12,13], without the need for assembling individual parts (Fig. 3b).

Once the new shape is fixed, it takes the role of the initial geometry of an un-actuated specimen, meaning that the material is limited to morphing into its initially synthesized, often basic geometry. For certain applications, it could be advantageous for the shape-change transition to be reversed. One straightforward method to realize this is to fuse several differently pre-programmed LCE pieces together, inducing bending or twisting

deformations due to the mismatching mechanical strains between the LCE layers[14,15] or other materials[16–18], but this requires a separate assembly step. Alternatively, a photomask can be used during the photocrosslinking process to selectively crosslink[19] or de-crosslink[20] an LCE sample, creating a patterned region of aligned and non-aligned material (Fig. 3c). Depending on the sample's thickness and the light penetration length, only the surface can be patterned, allowing for programming each side of the specimen differently. This way, the mechanical inhomogeneities are imprinted during the material synthesis itself.

The synthesized geometries are mostly limited to 2D strip-shaped approximations due to the synthesis methods of LCEs. These methods frequently involve significant amounts of solvents that need to be evaporated, or the thickness is constrained by the short light penetration lengths during photocuring. This makes any kind of mechanical stress programming very limited, especially with compressive stress, because of the usual small thickness of the samples. Therefore, LCEs need to rely on origami-like bending to morph into 3D objects. However, recent demonstrations show that with careful molding[21–23] or utilizing additive manufacturing techniques[24], LCEs can be prepared as robust 3D objects, ready for further shape programming (Fig. 3d). The progressively increased variety of initial and actuated shapes resulting from new synthesis methods can make LCEs much more applicable.

**Fig. 4 | New LCE functionalities originating from dynamic bonding. a** The introduction of dynamic bonds enhances LCEs with reprogrammability and the ability for material welding. Scale bars: 5 mm. Reproduced from ref. 29 with permission. © 2021 Wiley-VCH GmbH. **b** Dynamic bonds can also result in self-healing properties and enable reprocessing of LCEs. Reproduced from ref. 27 with permission. © 2021 Wiley-VCH GmbH.

This becomes even more evident with the introduction of exchangeable or dynamic bonds that enable reprogramming of the material. In general, dynamic bond exchange relies on the dissociation and recombination of chemical or physical bonds with the application of high temperature[25–31], light[32–36], or both[37–40]. A new mesogen configuration is mechanically instilled after the bonds are dissociated and subsequently imprinted into the material upon their recombination once the bond-exchanging stimulus is removed. For further insight, Saed et al. provide a comprehensive review on this topic[41]. Another way to achieve reprogrammability that was more recently applied to LCEs but is commonly used in shape-memory polymers[42], takes advantage of the persistent glassy state at room temperature in the LCE network. This state helps prevent strain relaxations by increasing the material hardness through the development of crystallites[43,44]. This enables a relatively straightforward approach to introduce a reprogrammable one-way or even reversible shape memory into LCE systems[45,46].

Although the original shape continues to be limited by the initial molding, the final shape can now be reprogrammed as needed using such methods. This is particularly noteworthy when reprogramming is not dependent on illumination, addressing concerns related to light penetration. As a result, it becomes possible to reshape larger specimens with relative ease. Moreover, numerous dynamic bond mechanisms offer reprocessability[27,28,37–39,47] and self-healing properties[28,29,38–40,47], allowing for reshaping the material into completely new initial forms or creating new shapes by welding prefabricated material pieces[47] (Fig. 4a, b). The practicality and multifunctional aspects of reprogrammable LCEs have thus generated significant attention for research in this direction.

## Patterning with surface alignment
The surface alignment approach is ideal for fabricating thin LCE sheets with well-defined mesogen orientations. Liquid crystal orientations tend to anchor to limiting surfaces, and their orientation at the surface influences the arrangement within the LC system due to self-ordering[48]. A well-established directional alignment is achieved by confining LCs between glass substrates coated with a polymer alignment layer[49] (Fig. 5a). LC molecules tend to align according to the direction of the microgrooves or molecular arrangement in the alignment layer, with the surface-induced layering preserved for several hundred micrometers[48,49]. This aligning mechanism is also used for imprinting mesogenic order in LCEs. In this process, the LCE synthesis mixture is introduced into an alignment cell, where the mesogen constituents reorient according to the alignment direction, and the ensuing order is preserved by further crosslinking the LCE mixture.

Basic preparation of the substrates involves rubbing the alignment polymer layer to produce linear microscopic grooves for mesogen orientation. Microchannels can also be directly written onto the alignment layer with photolithography, using a detailed photomask to pattern the alignment surface via photopolymerization[50] (Fig. 5b). Such patterned LCEs can exhibit highly complex shape changes that go beyond linear or bending actuations produced with conventional surface rubbing. Moreover, a

numerical method was developed to blueprint and fabricate alignment cells based on the desired shape change[51]. Even more sophisticated aligning patterns can be created by photoaligning the dye molecules in the alignment layer using polarized light[52–54] (Fig. 5c). This allows for an arbitrary image or pattern to be translated into a light polarization pattern that is duplicated onto the polymer alignment layer with precise photopolymerization[53,55], enabling high-resolution patterning of the LCE surface.

A photomask can be additionally employed during the photocuring stage of the already aligned LCE mixture to further pattern the shape response (Fig. 6a). For example, this method was used to create isotropic and aligned regions by localized curing at different temperatures[56] or initiate a localized photoreduction of gold salts to produce gold nanoparticle-rich areas for a patterned photothermal actuation[57]. Furthermore, by fabricating alignment cells with two different surface alignment directions, the mesogens are reoriented over the sample thickness, giving rise to controlled bending and twisting actuations[8,58–61] (Fig. 6b).

Voxelization of the director is thus possible using such methods, although there is still limited control of the mesogen orientation over the thickness, which depends on the aligning nature of the two limiting surfaces—each is fabricated to be exclusively planar or homeotropically aligning. Complete orientational control would therefore require additional surface modulation of the two kinds of photoaligning materials in a single alignment layer. Recently, the photoalignment process was further employed in conjunction with the two-photon polymerization method to produce micro-sized objects and microstructures[62–64], also with a non-linear mesogen arrangement[65] (Fig. 6c). Such methods might hold the potential to fully voxelize the mesogen alignment by selectively polymerizing individual voxels under different surface alignment conditions or, if upscaled, to further fabricate macro-sized systems through additive manufacturing. This however raises the question of whether such precise, microscopically programmed mesogen alignment is considered suitable for triggering macroscopic shape changes.

In general, the fabrication of patterned LCE sheets using alignment cells is a highly time-intensive process that typically requires specialized equipment to realize. Therefore, upscaling the production yield for practical purposes can be a challenging task. Nevertheless, the main attributes of surface-aligned LCEs lie in their two-dimensionality, making them particularly suitable for actuation with light and providing very fast actuation speeds[66–69], including snap-through[68], and oscillatory motions[57]. Concerning mechanical shape changes, the surface alignment technique is primarily better suited for fabricating smaller-sized actuators or shape-responsive surfaces.

## Alignment with EM orientational fields
Liquid crystals, when in the presence of electric or magnetic fields, tend to orient in the field direction due to their anisotropic properties, originating from their molecular structure (Fig. 7a). An electric field either exerts forces on the polarizable parts of the molecule or relies on the dielectric anisotropy

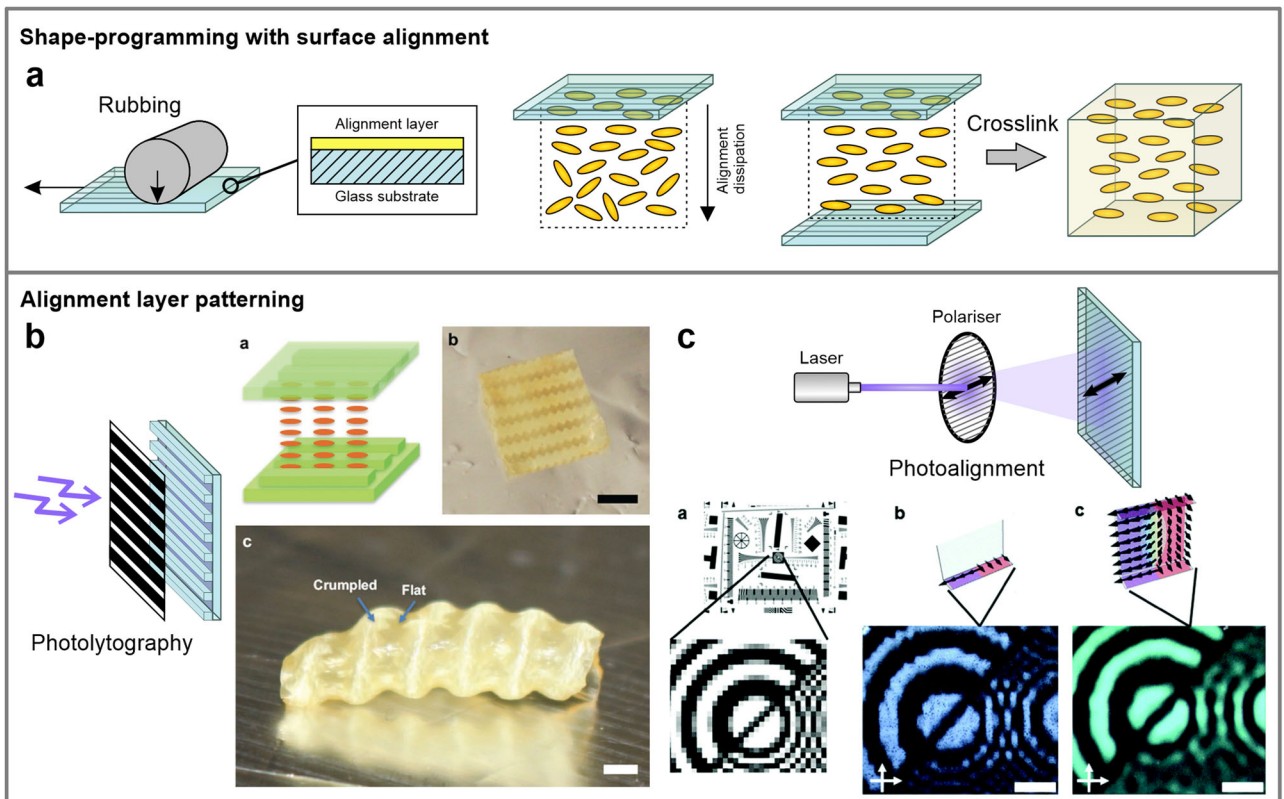

**Fig. 5 | Methods for patterning the alignment layer and shape-programming LCEs with surface alignment. a** A schematic of the alignment cell and mesogen ordering according to the direction of microscopic channels, produced by rubbing the alignment polymer layer on the glass cell. Alignment cells are utilized to produce ordered LCE thin films. **b** The alignment layer can be intricately patterned using photolithography. Scale bars: (b) 5 mm, (c) 1 mm. Reproduced from ref. 50 with permission. © 2016 WILEY-VCH Verlag GmbH & Co. KGaA, Weinheim. **c** With high-resolution photoalignment, an arbitrary image can be projected and inscribed into a new molecular arrangement in the alignment dye. Scale bars: 100 μm. Reproduced from ref. 54 with permission from the Royal Society of Chemistry.

of the LC molecules to induce torque, thereby reorienting LCs along the electric field direction[70]. Similarly, a magnetic field induces LC alignment owing to the anisotropic susceptibility of the LC molecules. Such aligning mechanisms are also employed for programming mesogen or LCE micro-particle orientation.

Electric field-assisted alignment was typically employed to establish homeotropic mesogenic order in thin LCEs[71–73], mostly for the interesting opto-mechanical effects or to tune mechanical properties of the material[74]. Shape programming with electric fields is essentially difficult to realize. High enough electric field strengths needed for mesogen orientation can only be achieved over short distances. Samples are therefore synthesized between glass plates with conductive ITO layers or patterned electrodes, typically reaching up to 100 μm in thickness (Fig. 7b). In this regard, it is more practical to exploit the alignment via surface anchoring for producing thin LCE sheets.

Therefore, the use of magnetic fields for shape programming can be a more viable option (Fig. 7c). If needed, strong magnets with several T can be easily procured and applied for the production of significantly large monodomain LCE samples and in substantial quantities[75] (Fig. 8a). Positioning magnets into different arrangements can also control the direction of the magnetic field near the magnet's surface, as demonstrated in producing a series of LCE microstructures with various morphing configurations[76].

The potential of EM fields lies in non-intrusive shape programming during the material synthesis itself, eliminating the need for post-processing, unlike the multi-step programming with mechanical stress. Despite the non-intrusive programmability, the size and shape of the final product remain restricted by the constraints of conventional synthesis techniques, such as molding. Moreover, modulation of the mesogen alignment with EM fields in larger samples is practically not feasible.

Several proposed techniques have combined EM fields with additive manufacturing techniques that can resolve the above-mentioned issues[77,78]. Two-photon polymerization was employed in the presence of an electric field to microprint structures with arbitrarily aligned LCE voxels[78] (Fig. 8b). The electric field is generated between two optically transparent electrodes that allows switching the electric field in any direction, while voxels are in-situ polymerized at the focus of the printing laser. Similarly, a printing device was constructed that enables switching the nematic director in a deposited LCE layer with a magnetic field and selectively photo-curing to create voxels with the desired in-plane mesogen orientation[77] (Fig. 8c). This allows a layer-by-layer fabrication of 3D soft devices from multiple LCE materials. However, both methods can only be used for the production of micro-sized architectures.

Despite the low production yields, both techniques demonstrate the utilization of EM fields for programming shape changes in LCEs. However, implementing strong EM fields into the production process can be challenging, given their interference with electronic equipment and field-sensitive components. In this respect, soft composites made from mono-domain LCE microparticles[46,79,80] can be beneficial in terms of shape programming, due to their increased susceptibility to EM fields. The enhanced susceptibility arises from the greater, collectively induced torque resulting from already aligned mesogens within the particles. Composite systems are shown to be considerably more convenient to mold into various shapes and sizes[46] (Fig. 8d), while further implementation into additive manufacturing techniques could enable voxelization of the particle alignment. As particle orientation does not rely on shear flow alignment, larger volumes of

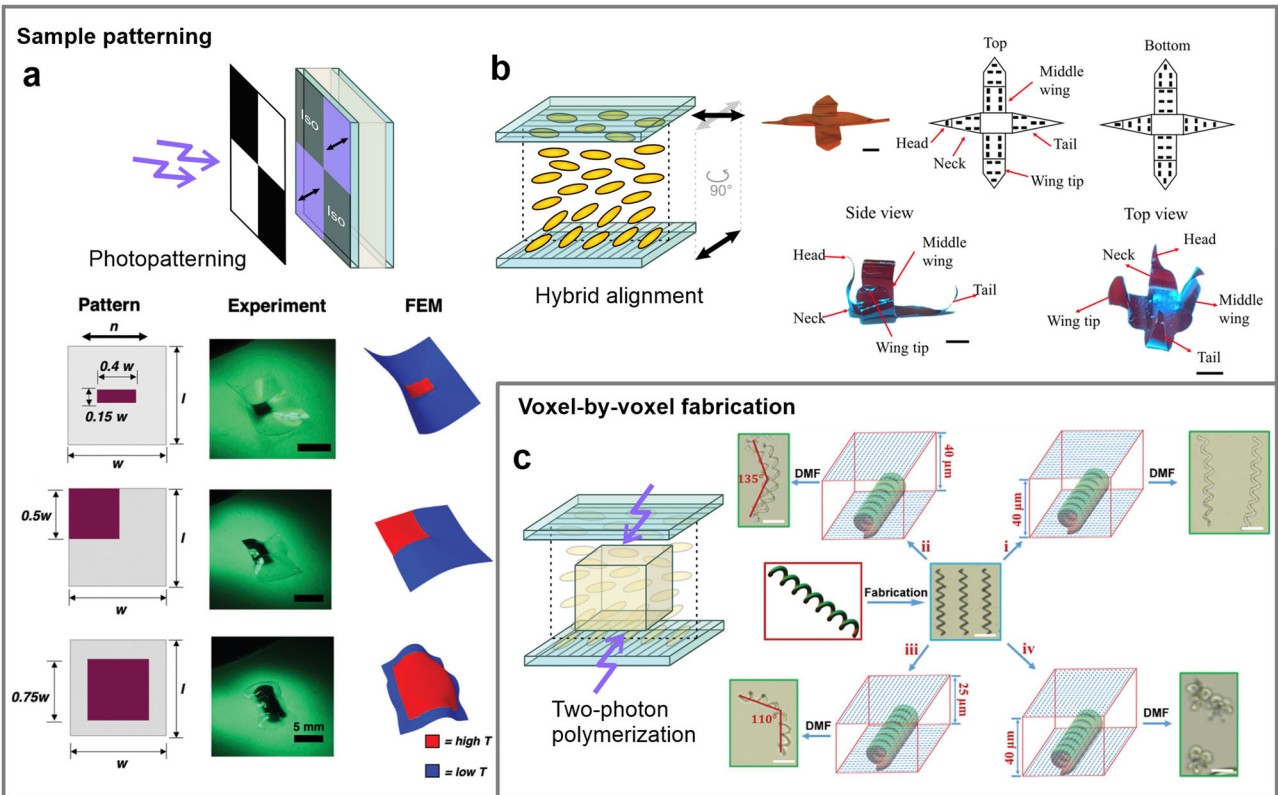

**Fig. 6 | Patterning of LCE thin-films and voxelization of mesogen alignment. a** Beside the alignment layer, the LCE synthesis mixture can be additionally patterned via photopatterning with a photomask. Reproduced from ref. 57 with permission. © 2020 WILEY-VCH Verlag GmbH & Co. KGaA, Weinheim. **b** The direction of mesogens can be altered over the thickness of the sample by using differently aligning and patterned layers. Scale bars: 2.5 mm. Reproduced from ref. 58. © 2022 The Authors. Advanced Intelligent Systems published by Wiley-VCH GmbH. **c** Two-photon polymerization allows for the fabrication of microstructures with voxelized mesogenic order. Scale bars: 50 μm. Reproduced from ref. 65. © 2020 The Authors. Published by Wiley-VCH GmbH.

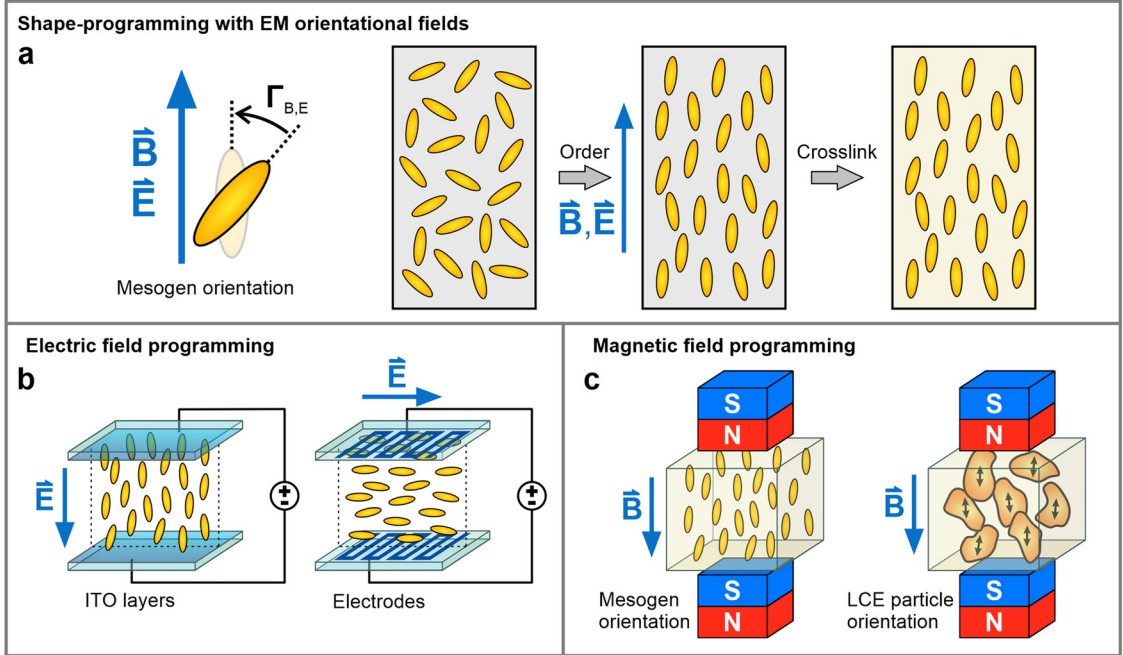

**Fig. 7 | LCE shape programming using an external magnetic or electric field. a** Applied EM fields induce mesogen reorientations against the field direction, providing a means to control mesogen alignment in the LCE pre-polymerized mixture. This alignment is then embedded into the material through the polymerization of the LCE network. **b** Vertical electric fields are generated between two parallel glass plates with conductive ITO layers. Patterned electrodes on the alignment cell can realize horizontal electric fields. **c** Strong magnetic fields have demonstrated the ability to induce mesogen or LCE particle alignment over greater lengths compared to electric fields.

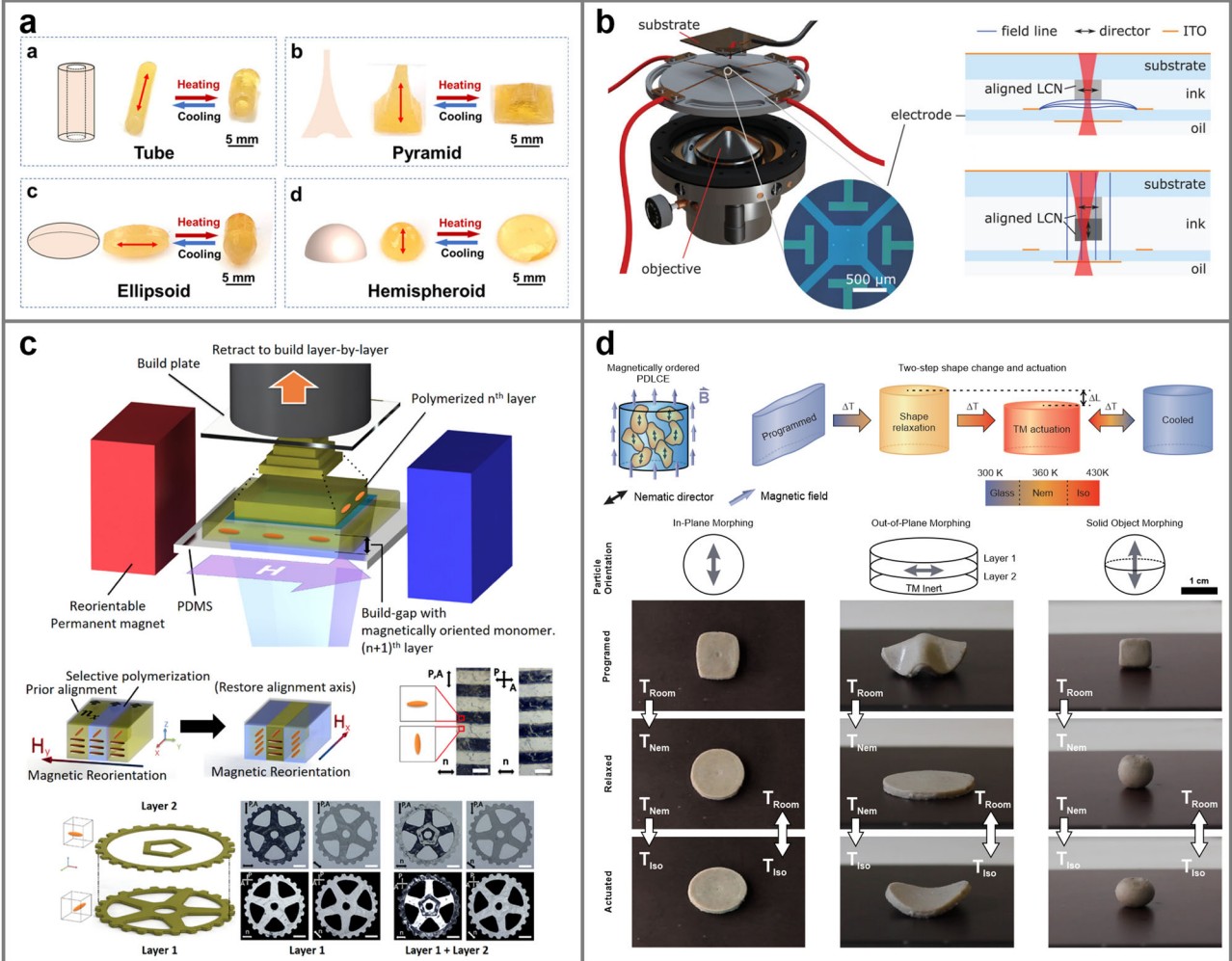

**Fig. 8 | Examples of shape programming methods with EM orientational fields. a** Magnetically ordered LCE systems have been molded into substantial-sized specimens. Reprinted with permission from ref. 75. Copyright © 2021 American Chemical Society. **b** Voxel control of mesogen alignment was achieved by two-photon crosslinking in the presence of an externally modulated electric field. Reproduced from ref. 78. © 2021 The Authors. Advanced Materials Technologies published by Wiley-VCH GmbH. **c** Selective photocuring of in-plane magnetically ordered LCE regions can be used to additively fabricate layered LCEs with patterned magnetic alignment. Scale bars: 1 mm. Reprinted with permission from ref. 77. Copyright © 2019 American Chemical Society. **d** A magnetic field was applied for the alignment of polymer-dispersed LCE particles to produce larger shape-morphing soft composites. Reproduced from ref. 46, licensed under CC BY 4.0.

magnetically orientable composite material could be deposited, potentially enhancing the production yield of such methods.

## Extrusion and shear-flow-induced alignment

Shear flow serves as a convenient aligning field that can be utilized in extrusion processes to fabricate monodomain LCE fibers. Typically, the LCE synthesis material is partially polymerized to form polymer chains that can be more easily aligned with the application of the flow. Upon extrusion, the chains are additionally crosslinked, mostly with photo-curing, to form the elastomer network and retain the alignment (Fig. 9a).

Extrusion can essentially be used solely for shaping the material. By depositing the LCE material onto a rotating spool and crosslinking, thick LCE fibers can be obtained[81]. Other methods like fiber-drawing[82], electrospinning[83,84] or even molding[22] were also applied to fabricate long LCE strands. These methods have the advantage of producing large quantities of LCE material, whereas the process still requires an additional mechanical programming step to achieve a monodomain state to ensure reversible shape changes. However, it has been demonstrated that efficient photo-crosslinking, coupled with the addition of carbon nanotubes to enhance mesogen alignment with flow, enables the extrusion of already crosslinked and well-aligned LCE fibers with large diameters[85] (Fig. 9b).

Another method involves gravity-induced alignment as part of the fabrication process[86], eliminating the need for additional mechanical programming of the fibers. Similarly, direct melt spinning[87] and melt electrowriting[88] techniques were applied to produce microfibers with well-established mesogen orientation, owing to the high shears and elongational forces present during the extrusion and deposition. In all cases, LCE fibers exhibit the ability to generate exceptionally large contractions while also displaying strong mechanical strength[22,81,82]. Although such fibers display only monotone contractions, there is often no necessity for intricate morphing configurations of the LCE material. Therefore, this method can be well-suited for implementation and integration into mechanical devices as active wires for inducing large actuations[81,82]. Furthermore, multi-component extrusion allows for the production of core-shell systems with advanced functions, such as conducting LCEs fibers with self-sensing capabilities[89] (Fig. 9c).

## Direct ink writing and additive manufacturing

Considerable research attention and advancements have been recently directed towards direct ink writing or so-called 4D printing, i.e., 3D printing of morphing materials with a predetermined shape change. Here, the extrusion process is combined with additive manufacturing techniques to

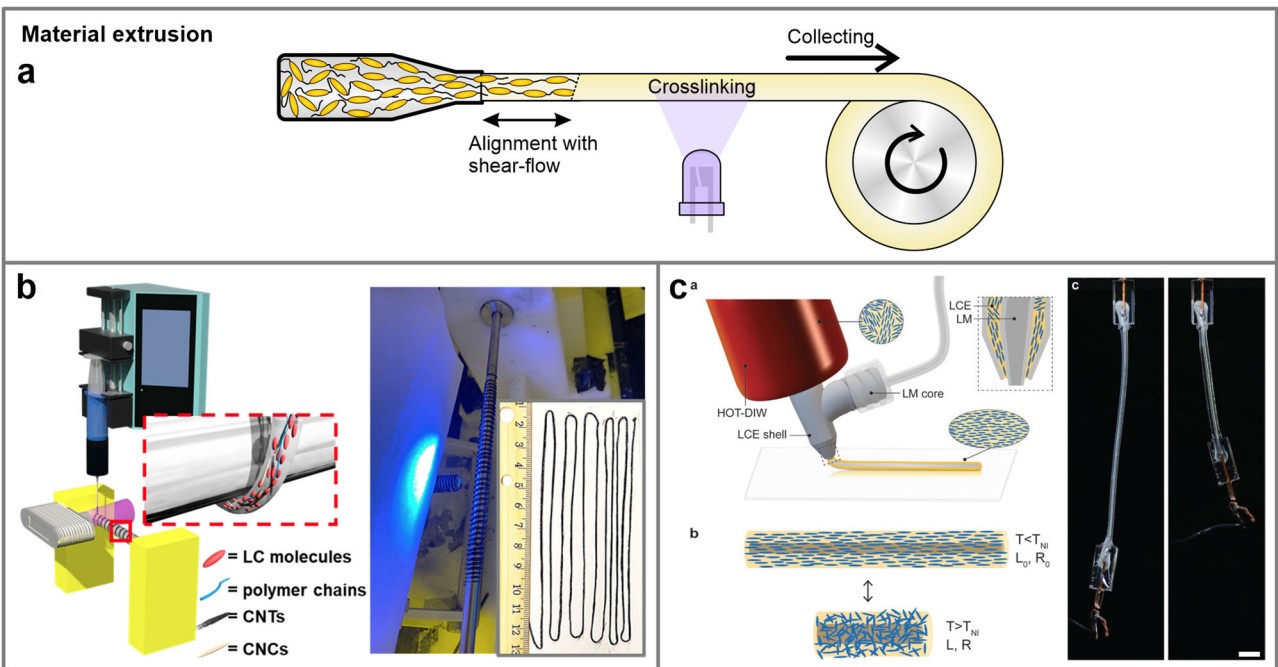

**Fig. 9 | Extrusion process for LCE shape programming. a** Extrusion of LCE material can utilize shear-flow to align LCE polymer chains. These chains are photocrosslinked after extrusion and collected to produce long, monodomain shape-memory fibers. **b** An example of long LCE strand production with incorporated carbon nanotubes for photothermal actuation. Reproduced from ref. 85. © 2021 The Authors. Advanced Materials Technologies published by Wiley-VCH GmbH. **c** Extrusion of aligned core-shell LCE fibers with a conductive liquid metal core. Scale bars: 5 mm. Reproduced with permission from ref. 89. © 2021 Wiley-VCH GmbH.

accurately deposit layers of extruded and aligned LCE material in a controlled manner (Fig. 10a). This process enables the creation of LCEs with intricate shapes and complex actuation behaviors[24,90–92]. These are predetermined by the printed mesogen alignment and can be further tuned by precisely controlling the direction and speed of the material deposition[93–96]. This method gained popularity because of its versatility to produce intricate structures that cannot be achieved using other methods, whereas the simultaneous control of mesogen alignment and shape within the process removes the necessity for post-processing and the application of external alignment fields.

While 4D printing offers the needed customization and precision in producing very detailed LCE artifacts[90,95], the modulation of the mesogen alignment with extrusion-based techniques is currently achievable only within the parallel plane of the deposition. In order to realize out-of-plane actuations, a variety of deposition or shape-programming methods has to be additionally introduced. These methods include depositing the material in spiral patterns[94,97,98] (Fig. 10b), forming a gradient material structure, for example, by introducing temperature gradients[99,100] or an uneven solvent evaporation rate[101] to alter the degree of mesogenic order in the deposited material (Fig. 10c), carefully adjusting and varying the printing parameters during deposition[93,96] (Fig. 10d), or even implementing a scratch post in the printing device to mechanically program the deposited material with compression and shear velocity[92]. Consequently, programmed LCE shapes are mostly constrained to 2D approximations, which need to rely on bending or folding mechanisms to transform into 3D geometries[94,97,99,102]. The absence of solid interiors of the actuated 3D shapes makes them harder to implement into processes where substantial stiffness of components is required.

From a production standpoint, additive manufacturing methods in general suffer from slow production speeds and scalability challenges compared to more established methods for polymer shaping and processing, such as extrusion or injection molding. It also relies on expensive equipment that usually is not suitable for printing LCE material, so additional modifications to the equipment and the LCE printing ink are essential parts of this method.

Original approaches to printing techniques and material synthesis already work towards the simplification of the printing process or introducing new features that rationalize the use of direct ink writing of LCEs. For instance, a system was developed for printing freestanding active material, where LCE ink is instantly cured after extrusion and can be additionally stretched with the nozzle to deposit and adhere between distanced surfaces[103], which can offer a direct assembly into multicomponent devices (Fig. 10e). An LCE material was also devised that utilizes dynamic hydrogen bonds as crosslinkers, which eliminates the requirement for photo-crosslinking and can be directly used with systems for printing thermoplastic polymer melts[104].

Further techniques for printing multimaterial systems have recently introduced that work towards removing the need for post-assembly by combining the fabrication routine into a single printing process, like the demonstrated printing method for simultaneous deposition of multiple LCE material[105] or producing LCEs with conductive traces or layers for Joule heating[106–108] (Fig. 10f). If combined with printable composite materials that exhibit additional material properties, for example, photothermal actuation[109], enhanced mechanical properties[110] or reprogramability[97,111], such techniques would pave the way towards 3D printing of multifunctional material that might enable direct manufacturing of smart devices.

## Discussion and outlook

Attaining full spatial control of mesogen alignment, where the mesogenic order would be freely and discretely oriented in arbitrary directions throughout the volume of a specimen, still presents a challenging goal in LCEs. Shape programming is oftentimes limited by the shape and size constraints posed by the material synthesis, where a 2D-like system became the standard LCE form. A planar mesogen alignment is therefore the prevalent option to achieve any significant shape changes, while the geometries are almost exclusively limited to morphing from or into flat 2D LCE sheets. Elaborate control of the director alignment in 3D objects is possible with non-uniform programing fields, which have been demonstrated to be efficiently controlled only over short distances, therefore near the surface of the

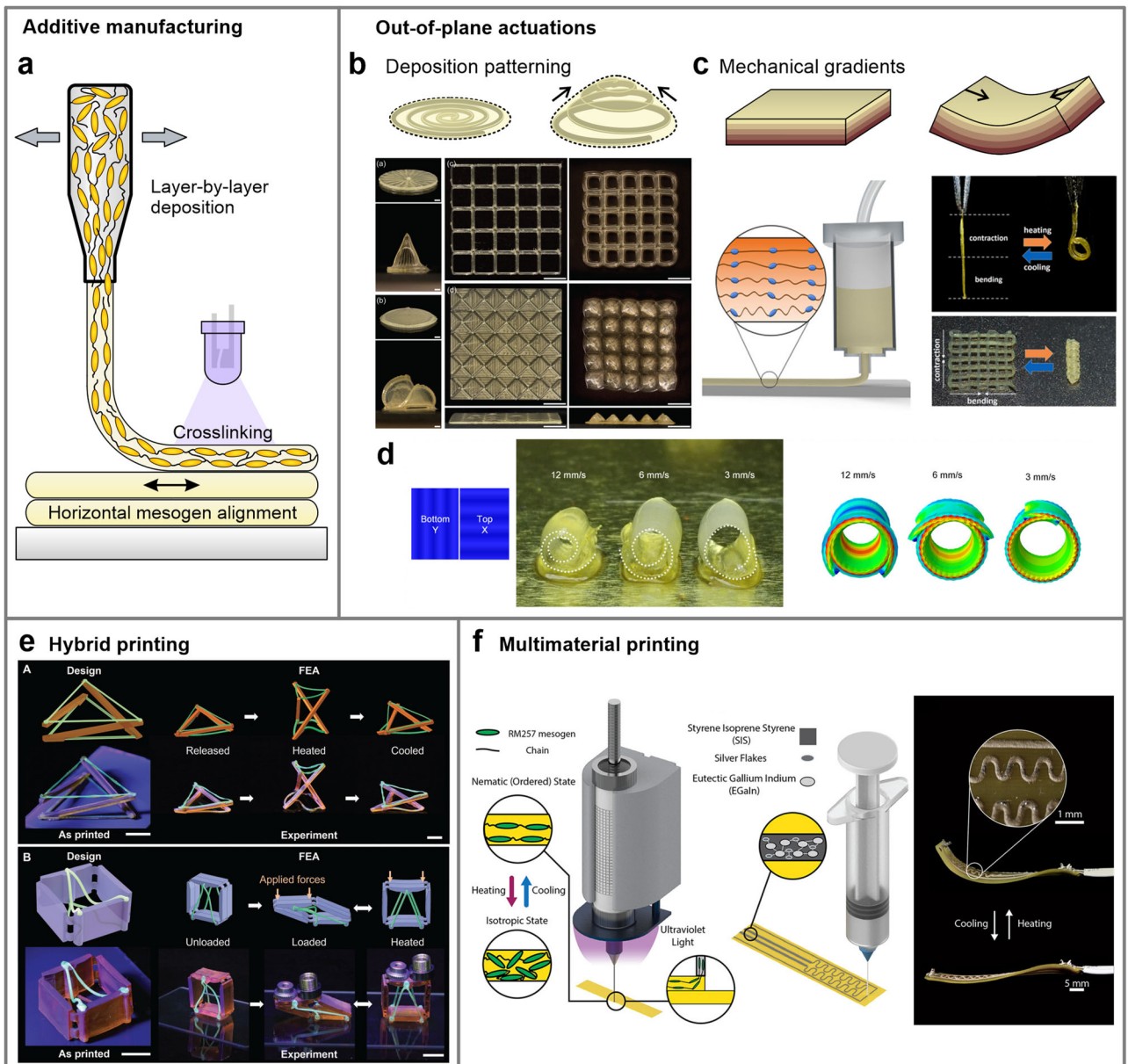

**Fig. 10 | Direct ink writing of LCE material. a** The extrusion method can be applied in additive manufacturing to deposit and cure aligned LCE fibers in layers, producing 3D specimens of elaborate shapes with an in-plane aligned and patterned mesogen alignment. **b** Out-of-plane actuations are attained by patterning the deposition in a spiral-like fashion. Scale bars: 5 mm. Reproduced with permission from ref. 95. © 2018 WILEY-VCH Verlag GmbH & Co. KGaA, Weinheim. Bending actuations are achieved by introducing additional mechanical gradients over the printed LCE's thickness. Examples demonstrate varying the mesogenic order over the deposited material's thickness with a temperature gradient (**c**, Reproduced with permission from ref. 99. Copyright © 2019 American Chemical Society.) or varying the printing speed to induce mismatching strains between the deposited layers (**d**, Reproduced with permission from ref. 93. Copyright © 2020 American Chemical Society.). **e** Printable free-standing LCE material enables hybrid printing of programmed LCE fibers in-between rigid components. Scale bars: 10 mm. Reproduced from ref. 103. © 2022 Wiley-VCH GmbH. **f** A multimaterial printing process is developed to fabricate LCEs with surface-deposited conductive traces for joule heating. Reproduced from ref. 106. Copyright © 2023 The Authors. Published by American Chemical Society.

material (e.g., through mechanical embossing) or in thin samples (e.g., with patterned alignment surfaces).

New additive manufacturing methods aim towards achieving full director modulation by combining shape programming with the voxel-by-voxel sample fabrication. If realized, such methods would enable complete freedom in both shape-forming and programming. As an example, it is worth mentioning an innovative additive manufacturing technique, the modified direct light processing printing method for printing voxelized LCE samples[112]. This method utilizes the sliding movement of the printing head to shear flow-align the mesogens in the LCE ink before selectively cross-linking specific voxels. However, the technique can achieve only parallel mesogen alignment, i.e., within the plane of the printing head movement. The application of external magnetic and electric fields therefore appears more promising for serving as non-intrusive orientational fields that could be dynamically applied in any direction during voxel-by-voxel material printing. Recent demonstrations have shown partial control of mesogen alignment and voxelization using external magnetic fields[77] (Fig. 8c) and complete control with fully modulated electric fields[78] (Fig. 8b). However, the demonstrated processes still exhibit extremely low material yields and slow production rates to apply for any kind of up-scaled manufacturing.

On the other hand, highly intricate shapes and deformations are not always needed in applications. Extrusion-based processes already offer a

practical and large yield approach for the production of LCE material with monotone actuation that can be used as active components for operation in conjunction with other mechanical parts[22,82,85,89]. LCE dopant-based composite systems[46,79,80,113], which can be easily molded without solvents, can offer a straightforward solution to seamlessly integrate soft actuating materials with solid components (like electronics or light sources) by molding around them. Such methods provide simplified manufacturing of bulk pieces that result in increased stiffness and work output needed for operation of larger mechanical devices. In this context, another promising aspect that could simplify LCE's incorporation into applications or the fabrication of soft shape-memory devices is the reprogrammability and reprocessability of dynamically crosslinked LCEs[41]. This characteristic makes them a versatile material that can be manufactured as on-demand programmable materials. Such LCEs could be prefabricated into standardized forms and later reprocessed and programmed into the required shapes.

Despite the various synthesis techniques and diverse properties, shape-memory materials have not yet been implemented in practical applications beyond the proof-of-concept demonstrations. In contrast to the production of conventional rubbers, LCE synthesis must involve an additional step for mesogen alignment or shape programming. Directly applying standard rubber production processes to LCEs is therefore not possible due to this requirement. Extensive modifications need to be performed to the fabrication method, which must in some way implement the required shape programming procedure, unless the LCE material itself is adapted for the specific process or generalized for broader implementation in production. As an example, a particularly promising prospect is the demonstrated application of dynamically bonded LCE material that does not require photo-crosslinking and behaves like a standard thermoplastic material. Therefore it can be applied with a non-modified nozzle for direct-ink printing of thermoplastics[104]. Development of such materials would make LCEs more accessible to everyday users or even hobbyists equipped with standard commercially available 3D printers.

However, unlike LCEs, conventional passive elastomer components continue to be reliably and inexpensively produced. Currently, there is no necessity to replace them with shape-morphing materials, at least for the existing operational purposes. The same applies to widely adopted active components, such as pneumatic or electromechanical actuators, which, considering actuation performance, still outclass LCEs in terms of speed, precision, and strength. The primary advantage of LCEs as actuators is in their softness, which makes them perfect for applications in geometrically complex or restricted environments, where material's shape adaptability is important. Therefore, the most obvious way to accelerate their utilization would be through further advances in newly emerging technologies and concepts that specifically exploit and take advantage of the unique soft characteristics of LCEs.

One such example is the field of soft robotics, which aims to replace the assembly of rigid parts with active elastic components to introduce adaptability for tasks impossible to achieve with a rigid skeleton. LCEs, as suitable candidate materials for such applications, have already been employed to demonstrate various functions for use in soft robotics, such as crawling[114–117], swimming[118,119] and gripping[66,120–122], among others[68,123–126]. However, the rudimentary performance of LCEs in their current state still requires them to be implemented as active elements working concurrently with other components in order to perform more complex tasks. The essential characteristic of LCEs, if they are to be employed as independent soft artificial limbs, such as resembling invertebrate appendages, would require a mechanism for localized stimulation to deform in the desired direction, as well as possess a solid interior. In this way, the robotic limb would be able to manipulate and interact with the surroundings without spatial constraints while also possessing high work density. This could potentially be achieved through multicomponent additive manufacturing methods or, more simplistically, by assembling preformed active LCE components alongside integrated driving mechanisms, such as conductive heating traces or light sources.

On the other hand, LCEs do not necessarily need to be employed for sophisticated applications that may extend beyond the current technological capabilities. One sector that can already significantly benefit from and drive further LCE implementation is the medical field, especially due to the material's biocompatibility[127–129]. Proposed applications taking advantage of the shape-memory properties of LCEs already range from drug delivery systems[130], active patches[131] to light-induced actuators for cardiac contraction assistance[132]. LCEs have also been utilized as active elastomer elements in multicomponent medical devices, such as self-deployable probes for neural recording[133].

Nevertheless, the application of LCEs as simple, soft, active components could be further extended with demonstrated applications in more consumer-close sectors. Such sectors could include the automotive industry, where active sealants and adaptive rubber surfaces might be some of the future uses of LCEs. Similarly, in the clothing industry, the already demonstrated woven fabrics from extruded LCE material[86,134] could pave the way to shape-adaptable clothing. This could also be particularly desirable for medical purposes, for example, in the creation of active compression garments. Comparable future sectors may also include smart architecture[135], where shape-programmed material is proposed to drive self-regulated processes for shading or ventilation, specifically beneficial for increasing energy efficiency of buildings, especially in combination with other smart materials, such as adaptive infrared absorbing components[136,137]. Additionally, integrating color-changing elements, such as mechanochromic materials[138,139] or light-emitting quantum dots[140] with shape-changing materials could serve as shape-interactive or haptic display indicators for wearable electronics.

Although specialized utilization is in general envisioned for smart materials, it would be beneficial to further demonstrate their operation within such everyday environments to induce exposure of the material to possible interests, which is also an important factor for further development and innovation. Ultimately, once such devices that fundamentally depend on soft shape-memory materials as vital components for their operation truly move beyond prototyping and demonstrate real-life applications, the demand for the already existing large variety of smart materials, including LCEs, is sure to follow.

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

## Acknowledgements

This work was supported by the Slovenian Research and Innovation Agency (ARIS), applied project L1-2607 and research program P1-0125.

## Competing interests

The author declares no competing interests.
