## [Peer Review File · Communications Chemistry]

Reviewers' comments:

Reviewer #1 (Remarks to the Author):

This article briefly reviews the commonly used shape programming methods, such as mechanical stress, surface arrangement, electromagnetic field, shear flow, etc., and discusses their advantages and limitations in preparing LCEs as programmable soft actuators. This manuscript also introduces some of the latest methods, such as dynamic bonding, reprogramming, two-photon polymerization, etc., to overcome some of the limitations of traditional techniques. The sentences are precise and accurate and the context is logical and convincing. However, the content of the article is not rich enough, and thus there remain some questions to be improved. I would recommend this manuscript for publication in Communications Chemistry after the revision. Here are some recommendations:

1. For the introduction part, I recommend adding a summary graph on shape-morphing LCEs by different orientation methods to indicate what the review talks about.
2. From Fig. 2 to Fig. 9, these graphs should keep a uniform format. For example, Fig. 5 figure has the frame to separate its sub-figures but Fig. 6 is not.
3. Please add the following references to enrich and support Fig. 2: [1] Materials Horizons, 2021, 8(9): 2475-2484; [2] Advanced Functional Materials, 2023: 2312068. [3] Advanced Functional Materials, 2022, 32(26): 2201884. [4] Angewandte Chemie International Edition, 2023, 62(25): e202304081. [5] Nano Today, 2022, 43: 101419;
4. In Fig. 3, please cite the following articles to enrich this part: [1] Nature materials, 2014, 13(1): 36-41. [2] Angewandte Chemie International Edition, 2023, 62(43): e202309402. [3] ACS applied materials & interfaces, 2017, 9(38): 33119-33128. [4] Advanced Functional Materials, 2023, 33(9): 2211914. [5] Nature Communications, 2023, 14(1): 7672.
5. In Fig. 8, please add the following references: [1] Nature Nanotechnology, 2022, 17(11): 1198-1205. [2] Advanced Materials, 2023, 35(14): 2209244. [3] Science Robotics, 2021, 6(57): eabi9704.
6. In Fig. 9, please update these cited references: [1] Matter, 2023, 6(4): 1278-1294. [2] Advanced Materials, 2023: 2307210. [3] Advanced Functional Materials, 2023: 2309019. [4] Angewandte Chemie International Edition, 2021, 60(10): 5536-5543.
7. In addition, the following references can be also cited, which would draw much more attentions of scientists and engineers from different backgrounds:
Nature Communications, 2023, 14, 3036; Materials Horizons, 2023, DOI: 10.1039/D3MH01386C; Angewandte Chemie International Edition, 2022, 61, e202211030; Advanced Materials, 2021, 33, 2004754; Materials Horizons, 2021, 8, 728-757.

Reviewer #2 (Remarks to the Author):

The authors attempt to give an overview of shape programming and preparation of liquid crystal elastomers, but unfortunately they have a poor understanding of the LCE literature. The article contains many incorrect and inaccurate statements. I do not recommend acceptance for publication.

1. "For instance, the polymerization of LCEs often relies on the use of significant amounts of solvents to

ensure the production of a homogeneous mixture and to serve as the reaction medium." - This is not true as many (probably most in the latest literature) LCEs and LCNs are made using reactive monomers and no solvent

2. "Furthermore, given that these reactions are conducted at elevated temperatures, an additional thermomechanical relaxation can be expected to occur upon cooling of the aligned specimen" This is also inaccurate as many LCE chemistries and reactions are conducted at room temperature.

3. "However, this synthesis process faces constraints due to the limited penetration length of light during curing, which restricts the achievable thickness in the synthesized LCE." Thick samples (hundreds of micrometers) have been made using photopolymerization. What exactly do the authors consider "limited penetration length"? It would be better if they specified a length or size limitation.

4. "The synthesized geometries are mostly limited to 2D strip-shaped approximations due to the synthesis methods of LCEs. These methods frequently involve significant amounts of solvents that need to be evaporated, or the thickness is constrained by the short light penetration lengths during photocuring" There are several examples of LCEs of sophisticated shapes either made by molding, additive manufacturing, or even using thin films to start with. The authors should state a thickness constraint (be more specific) if they want to point this out.

5. "The potential of EM fields lies in non-intrusive shape programming during the material synthesis itself, eliminating the need for post-processing, unlike the multi-step programming with mechanical stress. However, the process still relies on molding" It is unclear how processing using EM fields rely on molding or what exactly the authors mean by this statement.

"Full modulation of the mesogen alignment still presents a challenging goal to achieve in LCEs." What exactly do the authors mean by full modulation? LCEs with complex alignments throughout the samples are already possible.

"However, unlike LCEs, conventional 'passive' elastomer components continue to be reliably and inexpensively produced. Currently, there is no necessity to replace them with smart materials, at least for the existing operational purposes. The same applies to widely adopted 'active' components, such as pneumatic or electromechanical actuators. Considering actuation performance, these still outclass LCEs in terms of speed, precision, and strength." Very strange that the authors would state this. There are several needs for improvement of conventional elastomers, and if the authors knew of latest work they would be aware of LCEs for sensing biomedical applications touch sensors, etc. It is also difficult to know what the authors mean by conventional elastomers outclassing LCEs in terms of speed and precision - conventional LCEs do not change shape!

Reviewer #3 (Remarks to the Author):

This paper nicely reviewed the methods developed in the past for programming the shape of LCEs. The information will be very useful for the readers. The quality of the review paper will be further improved if the potential practical applications of the programmed LCEs can be discussed. I would like to recommend to accept the paper for publication with minor modifications.

Response to reviewer's comments

Reviewer #1:

We are thankful for the Reviewer's evaluation of the manuscript and given suggestions to enhance the text. Below are the listed corrections we made based on the proposed recommendations:

Comment 1: For the introduction part, I recommend adding a summary graph on shape-morphing LCEs by different orientation methods to indicate what the review talks about.

Author's response - A new Figure 1 has been made that schematically describes the LCE production challenges of shape-programming of LCEs and summarizes the main theme of the review article.

The added caption reads: LCE production challenges. The combination of forming and shape-programming methods is guided by geometrical and mesogen alignment limitations, impacting both material shaping and subsequent shape-change. Each LCE production method permits specific modes of actuation, thereby restricting the potential applications of the final material.

Comment 2: From Fig. 2 to Fig. 9, these graphs should keep a uniform format. For example, Fig. 5 figure has the frame to separate its sub-figures but Fig. 6 is not.

Author's response – Frames have been added to all the figures that contain multiple sub-figures.

Comment 3:

3. Please add the following references to enrich and support Fig. 2: [1] Materials Horizons, 2021, 8(9): 2475-2484; [2] Advanced Functional Materials, 2023: 2312068. [3] Advanced Functional Materials, 2022, 32(26): 2201884. [4] Angewandte Chemie International Edition, 2023, 62(25): e202304081. [5] Nano Today, 2022, 43: 101419;

4. In Fig. 3, please cite the following articles to enrich this part: [1] Nature materials, 2014, 13(1): 36-41. [2] Angewandte Chemie International Edition, 2023, 62(43): e202309402. [3] ACS applied materials & interfaces, 2017, 9(38): 33119-33128. [4] Advanced Functional Materials, 2023, 33(9): 2211914. [5] Nature Communications, 2023, 14(1): 7672.

5. In Fig. 8, please add the following references: [1] Nature Nanotechnology, 2022, 17(11): 1198-1205. [2] Advanced Materials, 2023, 35(14): 2209244. [3] Science Robotics, 2021, 6(57): eabi9704.

6. In Fig. 9, please update these cited references: [1] Matter, 2023, 6(4): 1278-1294. [2] Advanced Materials, 2023: 2307210. [3] Advanced Functional Materials, 2023: 2309019. [4] Angewandte Chemie International Edition, 2021, 60(10): 5536-5543.

7. In addition, the following references can be also cited, which would draw much more attentions of scientists and engineers from different backgrounds:

Nature Communications, 2023, 14, 3036; Materials Horizons, 2023, DOI: 10.1039/D3MH01386C; Angewandte Chemie International Edition, 2022, 61, e202211030; Advanced Materials, 2021, 33, 2004754; Materials Horizons, 2021, 8, 728-757.

Author's response – We have incorporated all suggested references, emphasizing the most significant ones directly in the text. The discussion section was further expanded by an outlook on the applicability of LCEs as requested by reviewer #3. The new text incorporates some of the suggested references. In addition, we have cited 15 more relevant works throughout the manuscript (references 9, 17, 18, 85, 115, 116, 117, 118, 119, 120, 121, 125, 128, 130, 139). Please find the suggested references numbers in the corrections below:

Corrections:

Point 3:

- [1] Materials Horizons, 2021, 8(9): 2475-2484; - **REF.NO. 62 – added to Fig.5**
- [2] Advanced Functional Materials, 2023: 2312068. – **REF.NO. 31 – added to Fig.3**
- [3] Advanced Functional Materials, 2022, 32(26): 2201884. – **REF.NO. 23**
- [4] Angewandte Chemie International Edition, 2023, 62(25): e202304081. – **REF.NO. 16**
- [5] Nano Today, 2022, 43: 101419; - **REF.NO. 7**

Point 4:

- [1] Nature materials, 2014, 13(1): 36-41. – **reference was already included – new REF.NO. 25**
- [2] Angewandte Chemie International Edition, 2023, 62(43): e202309402. - **REF.NO. 40**
- [3] ACS applied materials & interfaces, 2017, 9(38): 33119-33128. - **REF.NO. 41**
- [4] Advanced Functional Materials, 2023, 33(9): 2211914. - **REF.NO. 39**
- [5] Nature Communications, 2023, 14(1): 7672. – **this work was not included due to the different subject (hydrogels)**

Point 5:

Added new text on Page 12, Paragraph 2, Line 323:

Similarly, direct melt spinning⁸⁸ and melt electrowriting⁸⁹ techniques were applied to produce microfibers with well-established mesogen orientation, owing to the high shears and elongational forces present during the extrusion and deposition.

- [1] Nature Nanotechnology, 2022, 17(11): 1198-1205. - **REF.NO. 88**
- [2] Advanced Materials, 2023, 35(14): 2209244. - **REF.NO. 89**
- [3] Science Robotics, 2021, 6(57): eabi9704. - **REF.NO. 84**

Point 6:

Added new text on Page 16, Paragraph 5, Line 499:

Additionally, integrating color-changing elements, such as mechanochromic materials^{139,140} or light-emitting quantum dots¹⁴¹ with shape-changing materials could serve as shape-interactive or haptic display indicators for wearable electronics.

[1] Matter, 2023, 6(4): 1278-1294. - **REF.NO. 141**

[2] Advanced Materials, 2023: 2307210. - **REF.NO. 102**

[3] Advanced Functional Materials, 2023: 2309019. - **REF.NO. 112**

[4] Angewandte Chemie International Edition, 2021, 60(10): 5536-5543. - **REF.NO. 99**

Point 7:

References were included in the newly added text in the discussion section starting on Page 16.

Nature Communications, 2023, 14, 3036; - **REF.NO. 126**

Materials Horizons, 2023, DOI: 10.1039/D3MH01386C; - **REF.NO. 140**

Angewandte Chemie International Edition, 2022, 61, e202211030; - **REF.NO. 137**

Advanced Materials, 2021, 33, 2004754; - **REF.NO. 138**

Materials Horizons, 2021, 8, 728-757. - **REF.NO. 127**

Reviewer #2:

We thank the Reviewer for the evaluation of the draft, despite not receiving a positive assessment. While some concerns are a result of unclearly explained statements on our part, which might lead to incorrect conclusions, we also believe that some of the raised issues were taken out of context by the reviewer. We have addressed these issues with corrections to the text where deemed appropriate. The responses and corrections are provided below:

Comment 1: "For instance, the polymerization of LCEs often relies on the use of significant amounts of solvents to ensure the production of a homogeneous mixture and to serve as the reaction medium." - This is not true as many (probably most in the latest literature) LCEs and LCNs are made using reactive monomers and no solvent

Author's response – We do not agree with the Reviewer's remark regarding the statement being wrong. The text serves as an example and does not imply that all reactions include large amounts of solvents. The Reviewer themselves also state that not all LCEs, but many, are made using solvents. The remaining methods still rely on solvents for the reaction (as it also does in our laboratory, for instance), while there are many recent examples in the literature where solvents are used at least in the initial crosslinking step and are then removed prior to further processing. Here are just some newer examples of synthesis using solvents: [1] Adv. Funct. Mater. 2022, 32, 2201884; [2] ACS Appl. Polym. Mater. 2023, 5, 9, 7477–7484; [3] Adv. Mater. 2023, 35, 2304378; [4] Adv. Funct. Mater. 2023, 2307202; [5] Angew. Chem. Int. Ed. 2023, 62, e202304081.

Nevertheless, we indeed have in this particular statement focused on the two-step thermally induced polymerization of LCEs without providing this information. In order to emphasize that we are stating this as an example of the difficult relationship between the crosslinking method and the final shape, as discussed in the prior sentence to the quoted one by the Reviewer, we have changed the text to be more specific.

The new text reads:

'As an example, a typical two-step LCE synthesis via thermally induced polymerization often relies on the use of significant amounts of solvents, necessary to ensure the production of a homogeneous mixture and increase the mobility of the reactants, especially during the first partial crosslinking stage. Following the molding process, these solvents must be completely removed prior or after the completion of the final crosslinking reaction, leading to the deformation of the cured LCEs due to deswelling. Furthermore, given that these reactions are conducted at elevated temperatures, an additional thermomechanical relaxation might be expected to occur upon cooling of the aligned specimen. This hampers precise control over the programmed and actuated shapes, especially when aiming for the synthesis of more complex and precisely defined geometries.'

Comment 2: "Furthermore, given that these reactions are conducted at elevated temperatures, an additional thermomechanical relaxation can be expected to occur upon cooling of the aligned specimen" This is also inaccurate as many LCE chemistries and reactions are conducted at room temperature.

Author's response – This statement should be more accurate in the new context of the rewritten paragraph that pertains to the thermally induced polymerization. Upon further reading, two sentences after the highlighted one, we have already stated, and continue to do so in the revised manuscript, that LCE chemistries involving UV-triggered polymerization indeed eliminate problems associated with high temperatures. Nevertheless, even with photo-curing, LCE synthesis still relies on thermally-induced partial crosslinking, often in generous amounts of solvents, particularly with the intention of molding LCEs or preparing longer polymer chains for extrusion purposes. The exception would be thin films prepared between alignment plates. In this regard, we agree with the reviewer's remark that many chemistries are performed at room temperature, but surely not all. We have never stated otherwise.

Comment 3: "However, this synthesis process faces constraints due to the limited penetration length of light during curing, which restricts the achievable thickness in the synthesized LCE." Thick samples (hundreds of micrometers) have been made using photopolymerization. What exactly do the authors consider "limited penetration length"? It would be better if they specified a length or size limitation.

Author's response – Compared to photo-polymerization, thermal induced curing can yield samples in the macroscopic range of several centimetres or more. Light cured LCE samples, as the reviewer also notes, are limited to hundreds of micrometers, unless resorting to layering production techniques. This limitation is well-known in the field, not only for LCEs but also for elastomers in general. While our original statement only mentioned this limitation, we have rewritten the text to include a 'definition' of sample thickness, emphasizing that specimens resulting from thermal-induced curing are macroscopic 3D objects, as opposed to thin films typical of photo-curing.

The new text reads:

'However, this synthesis process faces constraints due to the limited penetration length of light during curing, which restricts the achievable thickness in the synthesized LCE to several hundred micrometers. It is therefore not possible to use conventional photopolymerization processes to mold

LCEs into bulk systems with a thickness comparable to other dimensions, typically in the centimeter range, unless relying on layering or additive manufacturing techniques.'

Comment 4: "The synthesized geometries are mostly limited to 2D strip-shaped approximations due to the synthesis methods of LCEs. These methods frequently involve significant amounts of solvents that need to be evaporated, or the thickness is constrained by the short light penetration lengths during photocuring" There are several examples of LCEs of sophisticated shapes either made by molding, additive manufacturing, or even using thin films to start with. The authors should state a thickness constraint (be more specific) if they want to point this out.

Author's response – This concern should now be addressed with the definition of thickness provided in the previous response. In addition, we have already acknowledged in the original text (beginning two sentences after the highlighted issue - page 4, paragraph 3, line 125) the point raised by the reviewer, namely that molding and additive manufacturing can be used for the production of more intricate shapes. It is worth noting that our specific focus is on the 3D aspect.

The original text reads:

'However, recent demonstrations show that with careful molding²¹⁻²³ or utilizing additive manufacturing techniques²⁴, LCEs can be prepared as robust 3D objects, ready for further shape programming (Fig. 3d).'

Moreover, the entire section focuses on mechanical stress programming. In this regard, only few works have been dedicated to producing stress-programmed LCEs with sophisticated 3D geometries, and we believe that this is sufficiently highlighted throughout the section.

Comment 5: "The potential of EM fields lies in non-intrusive shape programming during the material synthesis itself, eliminating the need for post-processing, unlike the multi-step programming with mechanical stress. However, the process still relies on molding" It is unclear how processing using EM fields rely on molding or what exactly the authors mean by this statement.

Author's response – We agree with the Reviewer that the sentence is unclear. We failed to explain that while EM fields serve as a non-intrusive shape-programming field that does not affect the final shape of the material during programming, such as with mechanical strain, the forming is performed through regular molding (e.g., cast molding or injection into glass cells), for which we already discussed the geometrical limitations in the earlier text. We then continue to explain how additive manufacturing can overcome these geometrical constraints in the following paragraph.

We have therefore revised the paragraph for better clarification (Page 10, paragraph 1, line 269):

'The potential of EM fields lies in non-intrusive shape programming during the material synthesis itself, eliminating the need for post-processing, unlike the multi-step programming with mechanical stress. Despite the non-intrusive programmability, the size and shape of the final product remain restricted by the constraints of conventional synthesis techniques, such as molding. Moreover, modulation of the mesogen alignment with EM fields in larger samples is practically not feasible.'

Comment 6: "Full modulation of the mesogen alignment still presents a challenging goal to achieve in LCEs." What exactly do the authors mean by full modulation? LCEs with complex alignments throughout the samples are already possible.

Author's response – To elaborate, while complex alignments have been demonstrated, mostly achieved with the use of alignment cells, these are typically in the form of repeated patterned

mesogen distributions in an in-plane fashion and/or formed together with a predetermined spatial gradient throughout the specimen's thickness (by utilizing mismatching alignment patterns, for instance). In all cases, precise control over the alignment is always only possible along two directions (typically in the plane of a strip-like specimen), while modulating the alignment in a similar discrete fashion over the thickness has not been achieved, but could perhaps be crudely realized with the use of spatially non-homogenous EM fields. Also, with direct ink writing, the control of the mesogen alignment can be performed only in the plane of deposition. Therefore, discrete mesogen alignment in all three directions over an LCE material has not been fully realized yet, at least to our knowledge, although some techniques show that it can be possible on the microscale, such as with two-photon polymerization under modulated EM fields, which paves the way towards voxel-by-voxel fabrication of arbitrary-aligned material units. In our opinion, this would demonstrate 'full modulation' of mesogen alignment. We agree that the sentence itself does not indeed explain this. We have therefore changed the sentence to omit the word 'full modulation' in order to prevent misunderstandings. The new sentence should now be clearer (Page 15, Paragraph 1, Line 409):

'Attaining full spatial control of mesogen alignment, where the mesogenic order would be freely and discretely oriented in arbitrary directions throughout the volume of a specimen, still presents a challenging goal in LCEs.'

Comment 7: "However, unlike LCEs, conventional 'passive' elastomer components continue to be reliably and inexpensively produced. Currently, there is no necessity to replace them with smart materials, at least for the existing operational purposes. The same applies to widely adopted 'active' components, such as pneumatic or electromechanical actuators. Considering actuation performance, these still outclass LCEs in terms of speed, precision, and strength." Very strange that the authors would state this. There are several needs for improvement of conventional elastomers, and if the authors knew of latest work they would be aware of LCEs for sensing biomedical applications touch sensors, etc. It is also difficult to know what the authors mean by conventional elastomers outclassing LCEs in terms of speed and precision - conventional LCEs do not change shape!

Author's response – In the context of this manuscript, which explores the shape-programming capabilities of LCEs and emphasizes their mechanical actuation for use as actuators, we further argue that there is currently no pressing need for shape-morphing materials in the existing elastomer or actuator industry. As stated, 'at least for the existing operational purposes'. This statement is supported by the limited applications beyond simple prototypes that, for the most part, solely utilize the 'actuating' aspect of LCEs rather than together with the elastic feature of the material, which makes such demonstrations perform subpar in comparison with other mechanical actuators. The lack of entrepreneurial interest also shows this trend, with only a few spin-offs focusing on other merits of LCEs rather than their role as soft actuators. While we acknowledge the potential of LCEs and benefits of 'improving' conventional elastomers, we argue, as also elaborated in the text, that widespread utilization or demand for LCEs will only emerge once practical applications or demonstrations are realized that rely on these materials for their functionality. Currently, even if we were to overlook all geometric constraints and complex production/synthesis procedures, smart materials remain primarily a novelty.

Furthermore, by encouraging a re-reading of the original text, it becomes apparent that the comment regarding 'outclassing performance' pertains to the previous sentence discussing pneumatic and electromechanical actuators, which are extensively utilized in various technological domains including robotics, haptics, and industrial machinery. These traditional actuators indeed significantly surpass LCEs in terms of performance. We have nevertheless changed the sentences in this paragraph to be more specific and avoid misinterpretations of the text.

The new paragraph reads :

'However, unlike LCEs, conventional 'passive' elastomer components continue to be reliably and inexpensively produced. Currently, there is no necessity to replace them with shape-morphing materials, at least for the existing operational purposes. The same applies to widely adopted 'active' components, such as pneumatic or electromechanical actuators, which, considering actuation performance, still outclass LCEs in terms of speed, precision, and strength. The primary advantage of LCEs as actuators is in their 'softness', which makes them perfect for applications in geometrically complex or restricted environments, where material's shape adaptability is important. Therefore, the most obvious way to accelerate their utilization would be through further advances in newly emerging technologies and concepts that specifically exploit and take advantage of the unique soft characteristics of LCEs.'

Reviewer #3:

We appreciate the positive evaluation of the manuscript by the reviewer. In response to their suggestion, we have expanded the 'Discussion and outlook' section of the paper to provide a more comprehensive discussion of the potential practical applications of LCEs and further expanded on our take on the potential transition of LCEs into real-life use. Specifically, we examine applications in soft robotics and discuss the requirements for LCEs or shape-changing materials to function as standalone artificial soft limbs. This is followed by demonstrated examples of use in the medical field and a discussion on the potential of integrating LCEs into more consumer-close sectors such as automotive, clothing, electronics, or smart buildings/architecture. Examples of already demonstrated prototypes and LCE actuators are included. The section is concluded with an emphasis on the importance of showcasing such uses in order to help with the utilization of LCEs into real-life applications. The extended text is too long to include in this response, but can be read in the manuscript, starting on page 16, paragraph 2, line 459.

REVIEWERS' COMMENTS:

Reviewer #1 (Remarks to the Author):

The authors have carefully addressed the raised issues in details, and I would like to recommend the manuscript for publication as its current form.

Reviewer #3 (Remarks to the Author):

I think the authors have adequately addressed my questions.